# Treatment of Wastewater Using Seaweed: A Review

**DOI:** 10.3390/ijerph15122851

**Published:** 2018-12-13

**Authors:** Nithiya Arumugam, Shreeshivadasan Chelliapan, Hesam Kamyab, Sathiabama Thirugnana, Norazli Othman, Noor Shawal Nasri

**Affiliations:** 1Department of Engineering, Razak Faculty of Technology and Informatics, Universiti Teknologi Malaysia, Jalan Sultan Yahya Petra, Kuala Lumpur 54100, Malaysia; nithiya85.a@gmail.com (N.A.); hesam_kamyab@yahoo.com (H.K.); sathiabama@utm.my (S.T.); norazli.kl@utm.my (N.O.); 2Sustainable Waste-To-Wealth, UTM-MPRC Institute for Oil & Gas, Resource Sustainability Research Alliance, Universiti Teknologi Malaysia, Johor Bahru, Johor 81310, Malaysia; noorshaw@utm.my

**Keywords:** adsorption, biosorption, seaweed, algae, wastewater

## Abstract

Inadequately treated or untreated wastewater greatly contribute to the release of unwanted toxic contaminants into water bodies. Some of these contaminants are persistent and bioaccumulative, becoming a great concern as they are released into the environment. Despite the abundance of wastewater treatment technologies, the adsorption method overall has proven to be an excellent way to treat wastewater from multiple industry sources. Because of its significant benefits, i.e., easy availability, handling, and higher efficiency with a low cost relative to other treatments, adsorption is opted as the best method to be used. However, biosorption using naturally found seaweeds has been proven to have promising results in removing pollutants, such as dyes from textile, paper, and the printing industry, nitrogen, and phosphorous and phenolic compounds, as well as heavy metals from various sources. Due to its ecofriendly nature together with the availability and inexpensiveness of raw materials, biosorption via seaweed has become an alternative to the existing technologies in removing these pollutants from wastewater effectively. In this article, the use of low-cost adsorbent (seaweed) for the removal of pollutants from wastewater has been reviewed. An extensive table summarises the applicability of seaweed in treating wastewater. Literature reported that the majority of research used simulated wastewater and minor attention has been given to biosorption using seaweed in the treatment of real wastewater.

## 1. Introduction

Algae is a diverse group of photosynthetic organisms ranging from unicellular (microalgae) to multicellular (macroalgae) forms inhabiting freshwater and marine environments [1]. Macroalgae, commonly known as seaweeds, are fast-growing organisms that resemble plants with some species able to grow up to 60 m in length [2]. Due to their varying intrinsic characteristics, seaweeds are used in a variety of industrial processes, including ecosystem balancing in mitigating eutrophication for nutrient management or as a bioremediation [3,4]; seaweed extract as a biobased fertiliser for crops [5,6]; anaerobically digested for the production of energy-rich biogas [7,8]; as an edible fresh food [9], etc. Particularly in Asia, seaweeds are generally consumed fresh or utilised for the production of industrially important phycocolloids [10]. Seaweed is also a source of natural polymer, which is incorporated into conventional plastic formulations to develop biodegradable plastics [11]. Below, the industrial applications of different types of seaweeds are summarised (Table 1). The astonishing multiple usage and contribution of seaweeds have increased their need, demand, and cultivation. They have also been traditionally used in domestic applications as a protein source, which can be incorporated into several value-added food products, medicinal usages for health benefits, food material for animals, savoury flavour source, etc., as shown in Table 2.

More recently, there has been a growing interest in using seaweeds as potential agents to treat wastewater via adsorption. The presence of sulphated polysaccharides in the cellular wall of macroalgae, primarily its fibril matrix and intercellular spaces, is the main reason for its high capacity to bind pollutants, such as trace metals. In fact, hydroxyl, sulphate, and carboxyl groups of the polysaccharides are strong ion exchangers; therefore, they are the important sites of complexation of metal cations [28]. These abovementioned chemical compositions of cell wall vary considerably among different types of seaweeds and the preferable one depends on its useful purpose.

## 2. Seaweed Application in Wastewater Treatment

In addition to contributing its usage in multiple industries, seaweed was largely explored and used in wastewater treatment as an adsorbent in order to replace the functional activated carbon. Wastewater is a by-product generated from any process or activity. It could be from manufacturing industries, factories, landfills, households, textile industries, petrochemical industries, aquaculture, agriculture, etc. Organic and inorganic pollution in these wastewaters is a common scenario. The presence of large quantities of organic compounds is defined as organic pollution and this is similar for inorganic pollution [29,30]. The sources of these organic compounds originate from domestic sewage, urban run-off, agriculture and aquaculture effluents, treatment plants, and industrial effluents, such as paper and pulp making and food processing. Pesticides, fertilisers, hydrocarbons, phenolic compounds, plasticisers, biphenyls, oils, greases, detergents, and pharmaceuticals are some of the common organic pollutants [31]. Benzene, toluene, ethylbenzene, *p*-xylene (BTEX), dyes, and chemicals are some of the examples of organic pollutants [32,33]. Heavy metal ions, arsenides, and fluorides are some of the usual inorganic toxic pollutants [34,35] present and sourced from industries, such as agriculture, paint manufacturing, etc. [36] With the high-standard requirements set by environmental regulations on wastewater discharge, wastewater treatment has gained attention worldwide [37]. There are several studies being conducted to treat different types of wastewater with different approaches and adsorption onto macroalgae is not something new. It focuses on many aspects, such as the removal of dye, chemical oxygen demand (COD), biological oxygen demand (BOD), phenols, heavy metals, etc. Very limited studies have actually focused on COD and BOD removal, carbon fixation, lipid production, total organic carbon (TOC), and turbidity [38,39] from wastewater using macroalgae since the majority focus on the removal of dyes, phenols, and heavy metals.

### 2.1. Removal of Nitrogen and Phosphorus

The presence of excessive inorganic nutrients, such as nitrogen and phosphorus, due to anthropogenic sources, causes eutrophication in water bodies [40], manifested in an increased frequency of harmful algal blooms [41] in due course and causing hypoxia [42]. Eutrophication has become the primary water quality issue for most of the freshwater and marine ecosystems, as it causes a decline in coral reef health and loss of coral reef communities, increases the incidence of fish kills, and also decreases water transparency [43]. These nutrients could eventually also be used to enhance the dense growth of economically valuable aquatic plant life, i.e., seaweeds [44]. The ecosystem balancing role of seaweed aquaculture is not something new [45]. Seaweed’s capability of storing a high concentration of nitrogen in its tissue is its most important ability [46]. This inorganic nutrient bioextraction characteristic of seaweed has received extensive attention from authorities and researchers. Table 3 below shows some of the studies carried out in order to eliminate inorganic nutrients for the past decade. Most of the listed seaweeds play the role of a biofilter in between wastewater and the pollutants.

Studies have shown an agreeable fit of experimental kinetic data to the pseudo-first order model, summarising that the biosorption process consists of extracellular and intracellular transfer. By all means, extracellular transfer includes the attraction of phosphate by active sites on the surface of biosorbent (physisorption) and chemical bonds (chemisorption), while intracellular transfer contains biotransformation and intracellular accumulation [56,57]. There is also supporting evidence on the data that best fit the pseudo-second-order model. A pseudo-second-order equation is used to describe chemisorption involving valency forces through sharing or exchange of biosorbent–adsorbate electrons. Thus, this supports that more than one step including chemisorptions may be involved in the process of phosphate biosorption onto seaweed [58].

### 2.2. Removal of Phenolic Compounds

Other than the removal of nutrients, the removal of phenolic compounds from wastewater is also a common state. The evolutionary production and usage of phenol start with its basic usage as an antiseptic and later on extends to the synthesis of dyes, building block of bioplastics, and petrochemical and pesticide chemical industries. Phenol is also a versatile precursor to a large array of drugs, especially aspirin, herbicides, perfumery, pharmaceuticals, and cosmetics, including sunscreens, hair colourings, and skin lightening [59]. The global phenol production had reached 7.8 million tonnes by 2001 and exceeded 8 million tonnes worldwide in 2008 [60]. It is an essential commodity chemical, which is utilised in the invention of several industrially significant chemicals and polymers, including bisphenol A, phenolic resins, etc. [61]. Thus, phenolic pollutants widely existed in wastewaters, discharged from chemical plants, cooking plants, or petroleum refineries [62]. It is enormously urgent to remove them from wastewater due to their toxicity, carcinogenicity, teratogenicity, and mutagenicity. Nevertheless, phenolic compounds are difficult to be treated just using the conventional treatment method [63]; still, biological treatment [64] and adsorption, especially using macroalgae, have also been approached by the wastewater treatment industry [65]. Phenolic compounds biosorption using seaweed has been described as hydrophobic and has donor–acceptor interactions [66,67,68]. A similar behaviour can be seen for the adsorption of chlorophenols on granular activated carbon and it was suggested that these compounds were adsorbed at the carbonyl oxygens on the surface of activated carbon as per the donor–acceptor complexation mechanism [67].

### 2.3. Removal of Dyes

For the removal of dissolved organic pollutants like dyes sourced from industries, such as textile, paper and pulp making, food colouring, and the cosmetics and carpet industries, adsorption treatments are used as high-quality treatment processes. These industries reportedly utilise large quantities of dyes which are toxic and even pose carcinogenic effects, as well as being hazardous to aquatic organisms and mammalian animals. Most of the dyes consumed in the textile industries are not biologically degradable, and they are resistant to aerobic digestion. Methylene blue and malachite green are the two most commonly found dyes in wastewater. Different types of low-cost adsorbents were researched and used for dye removal while also searching for more economical and easily available materials. Activated carbons (AC), such as olive-based [69] and karanj fruit hulls-based [70], as well as waste-based ACs, like cashew nut shell-based [71], *Ficus carica* bast fibre-based [72], *Daucus carota* plant-based [73], and *Salix psammophila* plant-based [74], are some examples of adsorbents used in removing methylene blue. *Limonia acidissima* shell-based [75], potato peel-based [76], and *Daucus carota* plant-based [73] ACs are some of the waste-based adsorbents used in removing malachite green dye. Table 4 below lists the common polluting dyes in wastewater, including methylene blue and malachite green. During the majority of the removal treatments, seaweed performed as an adsorbent and, to date, there are many ongoing studies using natural seaweeds. Looking into the adsorption of dyes by seaweed, it is mainly due to the presence of active functional groups, such as hydroxyl, carboxyl, carbonyl, amine, and sulfate, which participated in the biosorption process. Studies on kinetic reaction disclosed that data fitted well with the pseudo-second-order kinetic model, indicating that the chemisorption mechanism took part in removing dyes from wastewater [77].

### 2.4. Removal of Heavy Metals

Seaweed mostly used in wastewater treatment also reduces or removes toxic heavy metal contents [85]. The removal of heavy metals from wastewater is a huge environmental challenge today and it has drawn significant attention because of the heavy metals’ harmful nature to the environment and living organisms, especially when exceeding the regulatory standards [86]. Some heavy metals are toxic and carcinogenic, even at minute concentrations. They are non-biodegradable and can easily accumulate in living organisms [87]. Heavy metal build-up in soil and groundwater is a growing concern [88] and soil parent material (lithogenic source) and different anthropogenic sources are the sources of heavy metals in wastewater; metal smelter, paint industry, fertilisers, agricultural processes, leather tanning, electroplating, alloy and battery manufacturing, and other industrial waste materials disposal [89,90].

Chromium (Cr), Nickel (Ni), Copper (Cu), Arsenic (As), Cadmium (Cd), Mercury (Hg). and Lead (Pb) are the globally alarming heavy metals [91]. Pb is highly toxic to the kidney, nervous system, and reproductive system [92], whereas Hg is a neurotoxin that inhibits the enzymatic activities for normal neurotransmission and causes structural damages [93]. Environmental As exposure may also cause noncancer health effects, leading to the formation of tumours aside from cancer [94,95]. As is transported through water bodies into the environment, absorbed from soils to plants, and built up in many types of food crops and aquatic plants, threatening human health. Studies have proven that rice may be the possible primary source of inorganic As [96]. Environmental exposure to Cd is made possible by human activities, such as the manufacture of cement and construction materials, welding alloys, foundries, manufacturing steel and alloys, electroplating industry, lamps, mines, urban waste and industrial waste incineration, coal ash, tanneries, fertilisers, and wood preservatives [97].

Nickel toxicity towards plants could cause alterations in the germination as well as in the growth of roots, stems, and leaves. It also has deleterious effects on plant physiological processes, i.e., photosynthesis, water relations, and mineral nutrition. Metabolic reactions in plants are affected too, since nickel has the ability to generate reactive oxygen species that causes oxidative stress [98]. Studies have also proven liver and spleen injury, lung inflammation, and cardiac toxicity if nickel is exposed to animals [99]. Chromium is one of the toxic elements widely used in the industry, particularly in paints and metal platings as corrosion inhibitors, which eventually enter the water bodies via effluents from tanneries, textiles, electroplating, mining, dyeing, printing, photographic, and pharmaceutical industries [100]. The presence of chromium in excess causes genotoxicity and oxidative damage to cells [101,102].

There are other heavy metals that coexist and each heavy metal poses its own risk in nature towards both the environment and humans if it exceeds the allowable limits. Reliable methods are necessary for the removal of heavy metals from aquatic and wastewater. A great deal of attempts has been devoted to the effective removal of heavy metals from wastewater. To date, numerous treatment methods have been used for heavy metal decontamination, i.e., chemical precipitation, reverse osmosis, ion exchange, ultrafiltration, nanofiltration, coagulation, flocculation, flotation, electrodialysis, etc. However, these methods pose their own disadvantages when specifically compared to other methods. The adsorption process is economical, effective, simple, and versatile, and has become the most preferred treatment for heavy metals removal [103]. Knowing that fact, the high price and limited reusability are the key problems deterring the extensive application of a commonly used adsorbent, activated carbon [104]. In that perspective, biosorption has surfaced as a promising method with a high efficiency even at minute amounts, low cost, no additional nutrient requirements, easy handling, as well as zero detrimental effects towards the environment [105,106] Therefore, seaweed with the abovementioned criteria and together with its natural ecosystem balancing role has stimulated research conducted abundantly on varying types of seaweed to access the adsorbent performances and efficiencies in removing heavy metals in different wastewaters. Table 5 below shows the summary of seaweed generally used for heavy metals removal.

All three groups of seaweed (red, brown, and green seaweed) have been widely used in numerous wastewater treatments. There are both real wastewater and simulated wastewater tests for the adsorption ability of seaweed. From Table 5, it can be summarised that very limited studies have focuses on adsorption using real wastewater [111] while abundant studies have used the aqueous solution as well as simulated wastewater. It is evident that Pb, Zn, Cd, Ni, and Fe were the main elements focused on in most of the treatments. Ion exchange theory, where the predominant mechanism is involved in the sequestration of heavy metals present in the wastewater, was also the subject of studies. It was also noted that the sum of ions bound to biomass was similar to the sum of metals displaced from the biomass. In fact, the natural cation exchanging properties of seaweed were well explained due to the presence of functional groups, such as carboxylic and sulfonic groups, on the surface of all red, brown, and green seaweeds [119,120,121,122]. These abovementioned chemical compositions of cell wall vary considerably among different types of seaweeds and the key difference in the cell wall matrix explained the variance in the affinity of brown, green, and red seaweeds for metal biosorption [123].

## 3. Conclusions

The high number of uses of seaweeds documented has attracted the attention of many researchers and served to underline the importance of seaweeds in this period. The multiple functions and uses of seaweeds discussed above will stipulate continuous cultivation and supply of high-quality raw seaweed materials. It is foreseen that industries relying on seaweeds have the prospect to uplift the socioeconomics. Without doubt, we still need to divulge many opportunities from these unique yet resourceful species, as seaweeds are one of the most fascinating and complex living resources. This paper aimed to give an overall review of seaweeds in the wastewater treatment industry. Biosorption using seaweed is obviously a promising method with naturally existing raw material and a higher efficiency with a low-cost investment on the treatment. It is noticeable that very few studies used real wastewater for the treatment and most are simulated. Therefore, it is recommended that future studies focus more on treatment using real wastewater. Synthetic or simulated wastewater treatment with seaweed may have some setbacks, as they do not resemble the real wastewater characteristics, which are relatively complex in nature. Moreover, the results obtained from simulated wastewater may not be applicable to the real application of wastewater treatment plants (WWTP).

## Figures and Tables

**Table 1 ijerph-15-02851-t001:** Industrial applications of seaweed.

Industrial Applications	Seaweed	Type	References
Bio-oil production, which can be used as a combustion fuel for green electricity generation	*Saccharina japonica*	Brown seaweed	[12]
Seaweed extracts as a raw material for the synthesis of bioplastic film	*Gracilaria salicornia*	Red seaweed	[13]
Seaweed for circular nutrient (N and P) management to reduce eutrophication levels in the aquatic environment Resource for biobased fertiliser production	*Saccharina latissima*	Brown seaweed	[14]
Biogas production	*L. digitata and S. latissima*	Brown seaweed	[15]
Production of biochar for carbon sequestration and soil amelioration	*Ulva ohnoi*	Green seaweed	[16,17]
Source for biofuels	-Not provided-	-Not provided-	[18,19]
Bioenergy (methane) potential of seaweed as a promising seaweed bioenergy option	*Laminaria hyperborea*	Brown seaweed	[8]
As a multiproduct source for biotechnological, nutraceutical, and pharmaceutical applications	*Gracilaria gracilis*	Red seaweed	[20]
Agar from seaweed species widely used as a gelling, thickening, and stabilizing agent	-Not provided-	-Not provided-	[21]
Carrageenan used as a home remedy to cure coughs and colds	-Not provided-	Red seaweed	[21]
Edible fresh food	*Gracilaria spp.*	Red seaweed	[22]
Seaweed anaerobically digested for the production of energy-rich biogas (methane)	-Not provided-	-Not provided-	[17]
Development of biodegradable plastics incorporating natural polymers into conventional plastic formulations	*Gelidium robustum*	Red seaweed	[11]

**Table 2 ijerph-15-02851-t002:** Seaweed in domestic applications.

Domestic Applications	Seaweed	Type	References
Protein source for human nutrition	*Kappaphycus alvarezii*	Red seaweed	[23]
Flavour supplement and as a savoury flavour source for seafood products	*Gracilaria fisheri*	Red seaweed	[24]
Agar from seaweed is active in reducing blood sugar level	-Not provided-	Red seaweed	[25]
Carrageenan is used to make traditional medicinal teas and cough medicines to cure cold, bronchitis and chronic cough	*C. crispus and Mastocarpus stellatus*	Red seaweed	[25]
Carrageenan used to cure diarrhoea, constipation and dysentery			[25]
Alginate used to reduce cholesterol level, exerting anti-hypertension effect			[25]
Phlorotannins prevent obesity and obesity-related disorders	*Eisenia bicyclis*	Brown seaweed	[26]
Animal feed	*Sargassum* sp.	Brown seaweed	[27]

**Table 3 ijerph-15-02851-t003:** Wastewater treatment containing nitrogen and phosphorus using seaweed.

Seaweed	Type	Type of Wastewater	Studied Parameters	Treatment Conditions	Pollutants	Treatment Performance	References
*Gracilaria lemaneiformis*	Red seaweed	Aquaculture water (Bay water)	t = 1–35 d	Co-culture with the fish *Pseudosciaena crocea* Cage aquaculture Seawater with salinity of 26–29 (24–27 during low tide) Surface water T = 18.4–26.0 °C Surface water pH = 7.43–7.83 t = 20 d	Nitrogen and Phosphate	N = 21.0% P = 28.6%	[47]
*Gracilaria tikvahiae*	Red seaweed	Shrimp wastewater	t = 7–18 d	Co-cultured with Pacific white shrimp *Litopenaues vannamei* Salinity 30.4–34.8g/kg T = 18–33 °C pH = 7.4–7.9 t = 18 d	Nitrogen	N = 35% (Recovery in seaweed)	[48]
*Gracilaria chouae*	Red seaweed	Aquaculture water (Bay water)	t = 1–47 d	Co-cultured with the black sea bream *Sparus macrocephalus* Salinity of 28.33–31.07 T = 16.61–22.68 °C pH = 8.16–8.2 t = 28 d	Nitrogen and Phosphate	N = 41.2% (NO_3_–N = 37.76%, NO_2_–N = 36.99%, NH_4_–N = 29.27%) P = 46.2% (PO_4_–P = 40.64%)	[49]
*Ulva lactuca*	Green seaweed	Reject water from anaerobically digested sewage sludge	t = 1–18 d	Salinity of 20% from artificial seawater T = 15 °C pH = 7.9–8.9 t = 18 d	Nitrogen and Phosphorus	N = 22.7 mg N g DW^−1^ d^−1^ P = 2.7 mg P g DW^−1^ d^−1^	[50]
*Chondrus crispus*	Red seaweed	Finfish culture effluent	T = 6 and 13 °C	Land-based Atlantic halibut farm t = 28 d each trial	Nitrogen	Net N = 2.0 kgm^−2^ (at T = 6 and 13 °C)	[51]
*Palmaria palmata*			T = 6 and 16 °C			Net N = 2.0 kgm^−2^ (at T = 6 °C) Net N = 4.0 kgm^−2^ (at T = 16 °C)	
*Gracilaria vermiculophylla*	Red seaweed	Aquaculture effluents	t = 1 month each trial	Land-based pilot scale system Salinity of 30 ppm Mean T oscillates between 10.96 ± 0.19 °C and 20.17 ± 0.03 °C pH = 7.2–8.9 t = 1 month	Nitrogen	N = 40.54 ± 2.02 gm^−2^ month^−1^	[52]
*Gracilaria caudata*	Red seaweed	Aquaculture effluents	t = 72 h	Co-cultured with microcrustacean *Artemia franciscana* T = 28 °C Salinity = 35 PSU t = 72 h	Nitrogen and Phosphorus	NO_2_ = 100% NO_3_ = 72.4% DIN = 44.5% PO_4_ shows significant increase	[53]
*Gracilaria birdiae*	Red seaweed	Shrimp wastewater	t = 4 weeks	Salinity of 30.1–30.7 PSU T = 27.2–29.4 °C pH = 7.9–8.1 t = 4 weeks	Phosphate (PO_4_^3−^) and Nitrate (NO_3_^−^)	PO_4_^3−^ = 93.5% NO_3_^−^ = 100%	[54]
*Gracilaria caudata J. Agardh*	Red seaweed	Shrimp wastewater	t = 75 d	Co-cultured with *in-situ* shrimp pond Salinity of 33 PSU Mean T = 29 °C pH = 8.07–8.26 t = 4 h	Nitrogen and Phosphorus	NO_3_–N = 49.6% PO_4_–P = 12.3%	[55]

DW = dry weight; DIN = dissolved inorganic nitrogen.

**Table 4 ijerph-15-02851-t004:** Wastewater treatment in removing dyes using seaweed.

Seaweed	Type	Type of Wastewater	Studied Parameters	Treatment Conditions	Type of Dyes	Treatment Performance	References
*Ulva lactuca*	Green seaweed	Aqueous solution	T, pH, and t	T = 25 °C pH = 8.0 t = 150 min Biomass = 2.0 g Biomass size = 1.0 mm	Malachite Green	94.5% (T = 25 °C) 93.8% (pH = 8.0) 97% (t = 150 min)	[78]
*Sargassum crassifolium*	Brown seaweed					95.7 (T = 25 °C) 95.6% (pH = 8.0) 98% (t = 150 min)	
*Gracilaria corticata*	Red seaweed					93.3% (T = 25 °C) 92.5% (pH = 8.0) 96% (t = 150 min)	
*Nizamuddinia zanardinii*	Brown seaweed	Pure textile methylene blue solution	Dye concentration (10–240 mg/L)	pH = 6.5 dye concentration = 160 mg/L)	Methylene blue	565.96 mg/g (*Nizamuddinia zanardinii*) 77.18 mg/g (*Gracilaria parvispora)*	[79]
*Gracilaria parvispora*	Red seaweed						
*Nizamuddina zanardini*	Brown seaweed	Aqueous solution	pH, biomass (1–9 g/L), salinity (0.1–40 g/L NaCl), dye concentration (10–50 mg/L)	pH = 2.0 Biomass = 1 g/L Salinity = 40 g//L NaCl t = 90 min	Acid Black 1 (AB1)	58.05% (pH = 2.0) 92.1% (Biomass = 4 g/L) 72.24% (Salinity 40g/L NaCl) 23.37 mg/g maximum biosorption capacity (Dye = 50 mg/L, Biomass = 1 g/L, pH = 2)	[80]
*Nizamuddin zanardini*	Brown seaweed	Aqueous solution	Dye concentration (10, 30, and 50 mg/L), biomass (1, 3, and 5 g/L), pH (2, 4, and 6)	t = 90 min rpm = 130 T = 27 °C pH = 2.0 Biomass size = 160–250 µm	Acid Black 1 (AB1) dye (Amino acid staining diazo dye)	35.59% (Biomass = 5 g/L, dye = 10 mg/L) 99.27% (Biomass = 5 g/L, pH = 2)	[81]
*Sargassum glaucescens*						98.12% (Biomass = 5 g/L, pH = 2)	
*Caulerpa racemosa var. cylindracea*	Green seaweed	Aqueous solution	Dye concentration (5–100 mg/L), pH (3 and 11), biomass (0.1 and 2 g)	t = 90 min (Equilibrium) T = 18 °C pH = 7	Methylene blue	98% (Dye = 50 mg/L, Biomass = 2 g) 95% (Dye = 50 mg/L, T = 27 °C, pH = 11)	[82]
*Caulerpa lentillifera*	Green seaweed	Aqueous solution	Biomass (0.5–2 g)	Particle size ≤ 20 µm T = 25 °C t = 1 h rpm = 130 pH = 7 ± 0.5	Astrazon^®^ Blue FGRL (AB), Astrazon^®^ Red GTLN (AR), and Methylene blue	Methylene Blue = 417 mg/g (Biomass = 0.5 g)	[83]
*Sargassum muticum*	Brown seaweed	Aqueous solution	pH (1–10), dye concentration (50–500 mg/dm^3^), t	Adsorbent size = 0.5–1 mm Biomass = 0.125 g rpm = 175 T = 25 °C t = 4 h pH = 4 Dye concentration = 50 mg/dm^3^	Methylene blue	97.4% (Treated with CaCl_2_) 98.2% (Treated with HCl) 98.0% (Treated with H_2_CO)	[84]

**Table 5 ijerph-15-02851-t005:** Summary of seaweed used for heavy metals removal.

Seaweed	Type	Type of Wastewater	Studied Parameters	Treatment Conditions	Heavy Metals	Treatment Performance	References
*Sargassum* sp.	Brown seaweed	Synthetic wastewater	Biomass t pH rpm Ions concentration	t = 60 min T = 25 °C Biomass size = 200 mesh Cd^2+^ Biomass = 0.5 g pH = 4 rpm = 150 Ions concentration = 5 mg/L Zn^2+^ Biomass = 1 g pH = 3 rpm = 200 Ions concentration = 5 mg/L	Cd^2+^ and Zn^2+^	Cd^2+^ = 95.3% (acid treated) Zn^2+^ = 90.3% (acid treated)	[107]
*Sargassum* sp.	Brown seaweed	Simulated wastewater	Ions concentration = 0–7 mmol/L)	Adsorbent size = 2.2 mm Adsorbent = 0.1 g T = 30 °C rpm = 150 pH = 5 t = 4 h (Ni^2+^) and 6 h (Cu^2+^)	Ni^2+^ and Cu^2+^	Cu^2+^ = 2.06 mmol/g Ni^2+^ = 1.69 mmol/g	[108]
*Ulva rigida*	Green seaweed	Simulated wastewater	With raw and chemically treated seaweeds pH = 2–7	T = 20 °C Ion concentration = 25 mg/L Adsorbent = 0.5 g Adsorbent size = 0.5 cm t = 5 h rpm = 180	As^3+^, As^5+^, Sb^3+^, Se^4+^ and Se^6+^	Se^4+^ = 0.5 mg/g (pH = 2–4) Se^6+^ = 0.2 mg/g (pH = 2–3) In raw forms, showed limited perspectives for arsenic removal but can be promising for selenium and especially antimony	[109]
*Sargassum filipendula*	Brown seaweed	Simulated wastewater	t = 720 min	Adsorbent size = 0.737 mm Adsorbent = 2 mg/L T = 25 °C rpm = 180 t = 24 h Ion concentration = 1 mmol/L pH = 3.5	Ag^+^, Cd^2+^, Cr^3+^, Cu^2+^, Ni^2+^, Pb^2+^ and Zn^2+^	Ag^+^ = 33.62% Cd^2+^ = 78.03% Cr^3+^ = 72.8% Cu^2+^ = 69.05% Ni^2+^ = 32.74% Pb^2+^ = 56.19% Zn^2+^ = 44.21%	[110]
*Gracilaria* sp.	Red seaweed	Landfill leachate	Gel/Adsorbent concentration = 10, 20, 50, and 100 mg/L t = 10 d	pH = 8	As, Fe, Ni, and Cd	Fe = 100% (t = 1, Ion = 10 mg/L) Cd = 100% (t = 5 d, Ion = 100 mg/L) As = 100% (t = 5 d, Ion = 50 mg/L) Ni = 98% (t = 10 d, Ion = 50 mg/L)	[111]
*Sargassum hystrix*	Brown seaweed	Simulated wastewater	Adsorbent = 0.5–10 g/L t = 3–120 min Ion concentration = 0.5–100 mg/L	Ion concentration = 10 mg/L Adsorbent = 10 g/L t = 120 min	Mn^2+^	85.6%	[112]
*Sargassum filipendula*	Brown seaweed	Simulated wastewater	T pH Adsorbent Ion Concentration	T = 34.8 °C pH = 4.99 Ion concentration = 152.10 mg/L Adsorbent = 0.49 g/L	Pb^2+^	96%	[113]
*Sargassum muticum*	Brown seaweed	Simulated wastewater	pH = 2, 3, 4, and 5 With raw and protonated seaweed	Adsorbent size = 5 mm Ion concentration = 10 mg/L Adsorbent = 100 mg rpm = 200 T = 23 °C t = 6 h	Sb^3+^	3.5 mg/g (pH = 5, protonated *Sargassum muticum*) 3.4 mg/g (pH = 4, raw *Sargassum muticum*) *Sargassum muticum* shows significant removal efficiency than *Aschophyllum nodosum*.	[114]
*Aschophyllum nodosum*	Brown seaweed	Simulated wastewater					
*Osmundea pinnatifida*	Red seaweed	Simulated wastewater	pH = 2–9 t = 0–3 h Biomass = 5–80 g/L Ion concentration = 50–400 mg/L	pH = 5 t = 60 min Biomass = 1 g Biomass size = 0.5 mm Ion concentration T = 25 °C rpm = 500	Cu^2+^ and Cd^2+^	Cd^2+^ = 57.29% (pH = 5, biomass = 20 g/L, ion concentration = 100 mg/L, t = 180 min) Cu^2+^ = 50.89% (pH = 5, biomass = 20 g/L, ion concentration = 100 mg/L, t = 180 min) Cd^2+^ = 62.9% (pH = 5, biomass = 20 g/L, ion concentration = 100 mg/L, t = 60 min) Cu^2+^ = 69.15% (pH = 5, biomass = 20 g/L, ion concentration = 100 mg/L, t = 60 min) Cd^2+^ = 75.36% (pH = 5, biomass = 20 g/L, ion concentration = 100 mg/L, t = 60 min) Cu^2+^ = 70.22% (pH = 5, biomass = 20 g/L, ion concentration = 100 mg/L, t = 60 min) Cd^2+^ = 75.84% (pH = 5, biomass = 20 g/L, ion concentration = 50 mg/L, t = 60 min) Cu^2+^ = 71.64% (pH = 5, biomass = 20 g/L, ion concentration = 50 mg/L, t = 60 min)	[115]
*Sargassum ilicifolium*	Brown seaweed	Simulated wastewater	pH = 3–5 Ion concentration = 20–200 mg/L Adsorbent = 0.2–0.8 g/L T = 20, 25, and 30 °C	pH = 3.7 Adsorbent = 0.2 g/L Ion concentration = 200 mg/L T = 25 °C t = 2 h	Pb^2+^	195 ± 3.3 mg/g	[116]
*Chondracanthus chamissoi*	Red seaweed	Aqueous solution	pH = 2–5.5 for Pb(II), 2–7 for Cd(II)	pH = 4 Adsorbent = 20 mg Ion concentration = 70 mg/L t = 48 h	Pb(II) and Cd(II)	Pb(II) = 1.37 mmol/g Cd(II) = 0.76 mmol/g	[117]
*Ceramium virgatum*	Red seaweed	Simulated wastewater	pH = 2–8 t = 0–120 min Biomass = 1–40 g/L T = 20, 30, 40, and 50 °C	Biomass size = 0.5 mm Biomass = 10 g/L rpm = 100 pH = 5 t = 60 min T = 20 °C	Cd^2+^	96% (pH = 5, Ion concentration = 10 mg/L, T = 20 °C) 97% (pH = 5, Ion concentration = 10 mg/L, T = 20 °C, biomass = 10 g/L, t = 60 min)	[118]

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
