# Peer review of "Treatment of Wastewater Using Seaweed: A Review"

_ijerph, 2018, doi:10.3390/ijerph15122851_

Round 1
Reviewer 1 Report
The author has addressed all the comments accordingly. This manuscript can be accepted for publication
Author Response
We appreciate the reviewer's comments and happy to receive the good news.
Reviewer 2 Report
I will start by saying that the authors have made substantial improvements to provide clarity to the manuscript. The tables have been worked thoroughly and now contribute towards the purpose of the paper. Still, the manuscript is not ready for publication. Based on the rebuttal letter, I partially understand why the authors opted to disregard some of the changes suggested.
Unfortunately, this contribution still describes a few aspects inaccurately. For example, the classification of seaweed as red, brown, and green do not resemble any phylogenetic association as stated in Line 38. This grouping is totally artificial. Also, only a limited number of species can grow up to 60m in length and this occurs only in temperate marine environments.
I read your manuscript carefully and still feel that there is a lot of information that those not help the aim of the paper. Again, in your rebuttal letter you explain why adding this much detail, so maybe shortening this information will still add the desired background without introducing “noise” to the message you are trying to convey.
Here is my suggestion:
Line 31: Algae is a diverse group of photosynthetic organisms ranging from unicellular (microalgae) to multicellular (macroalgae) forms inhabiting freshwater and marine environments (reference). Macroalga commonly known as seaweeds are fast growing organisms that resemble plants with some species able to grow up to 60m in length (reference). Due to their varying intrinsic characteristics, seaweeds are used in a variety of industrial processes including …… (Table 1). They have also been traditionally used in domestic applications such as….. (Table X).
Table X : you could construct another table with household applications, similar to what you did in for table 1.
More recently, there has been a growing interest in using seaweeds as potential agents to treat wastewaters via absorption (refs). Continue with Line 110: “the presence of sulphate…
The purpose of your manuscript is providing a review of how seaweeds are being used and can be used to treat wastewaters. Synthesizing the information as suggested brings the reader to the main point of your contribution instead of having to read about the many other uses of seaweeds.
Going back to the title: I suggest eliminating “naturally found”. “Treatment of wastewater using seaweed: A review” integrates all options of seaweed acquisition without eliminating any source. In your conclusion, you can strengthen the fact that seaweed aquaculture is not a requirement. Also in your conclusion, you mention a key aspect: There is lack of research assessing seaweed potential outside small-experimental scale, or what you call "real" wastewater. It will be a good idea to emphasize this.
Other minor comments:
1) Keep consistent with using seaweed. In your text, you interchange algae with seaweed but in your first sentences, you already mention algae also includes microalgae, which is not on the scope of your document.
2) Line 254: replace” types” with “groups”, replace “algae” with “seaweed”
3) Line 265: avoid using subjective descriptors: “incredible” and “surprised” can be changed. What is incredible and surprising to one may not be to other.
4) I will strongly suggest reading the following document as it will contribute to your case:
The Bioremediation Potential of Seaweeds: Recycling Nitrogen, Phosphorus, and Other Waste Products
5) Abstract:
Line 13: you can eliminate the first "and"
Line 15: adsorption methods by? if overall, please include the word" overall".
Tables: Eliminate the column "mechanisms" as they are all via absorption.
Author Response
We appreciate the reviewer's comments. Please find attached file.

This manuscript is a resubmission of an earlier submission. The following is a list of the peer review reports and author responses from that submission.
Round 1
Reviewer 1 Report
Authors tried to summered the application of seaweed in the treatment of wastewater containing various pollutions (e.g., nitrogen, phosphorous, phenolic compounds, dyes, and heavy metals). Dozens of references were listed in 3 tables, indeed. However, more useful information should be clearly provided, including the efficiency of treatment, sorption capacity and affinity, and possible mechanisms. It will be more understandable and convincing, if the authors change the structure of manuscript. And the written of English should also be promoted.
Here are some detail comments and suggestions in the following:
(1) the Introduction section: Authors spent too much on the applications of seaweeds in various industries (paragraph 2 and 4, Table 1), however, the theme of this manuscript is the application in wastewater treatment. I suggest that authors cut down the complicated and overlapped texts. Table 1 is enough. In addition, the structure of Table 1 can be slightly modified. I think it would be better to change the “industrial applications” to the first column. In the introduction, the wastewater pollutants and treatment methods should also briefly discussed.
(2) the “seaweed compositions” part. It would be better if the compositions are related to the application in wastewater treatment or biosorption of pollutants.
(3) in the “wastewater pollutants” part.
First of all, the subtitle should change to “application in wastewater treatment”.
Moreover, it seems that the descriptions in all 3.1, 3.2, and 3.3 depict the risks of these pollutants and other treatment methods. However, the treatment by seaweed is merely mentioned. In this part, the interaction of seaweed with pollutants, underlying mechanisms, as well as comparison with other methods, I think, are usually the main topic. I suggest that authors rewrite this part, which is the most important part of this review.
In 3.1, I don’t think it’s suitable to discuss phenolic compounds together with nitrogen and phosphorous.
For the 3 main tables (Table 2, 3, 4), I think it would be much better if authors added information such as removal efficiency, possible mechanisms, system conditions, sorption parameters.
Reviewer 2 Report
In this contribution authors present a review of what seems to be a nice proposal to treat wastewaters.
Although it has potential for publication, there are several key aspects that need to be addressed before I can endorse the document.
As a general comment, there are multiple typos that need to be corrected. Also, English in general needs to be improved. There are multiple paragraphs where words are missing to add meaning to the sentences. Other issues include deleting extra spaces, for example in line 20 after the word effectively or adding a space between words, for example line 10 after the period.
Title: At first, I thought the manuscript was based on collecting drifting seaweeds or seaweeds already washed on shore and not necessarily collected from wild populations. As I read, there seems to be multiple sources. I suggest adjusting the title.
I guess my confusion arouse because plenty of the examples used in the document describe processes via some sort of seaweed cultivation.
Abstract:
Line 11: change “which becomes” use “becoming”. The same line reads awkward to me: becoming a concern prior to the release in the environment? I will suggest rephrasing.
Line 12: there seems to be a word missing before “absorption”.
Introduction and seaweed composition sections:
Besides grammatical errors, there are huge inaccuracies in the information provided in this section.
Just to mention one: Line 31 states that macroalgae inhabit marine environments, which is correct but the sentence is written in such way that expresses microalgae inhabit fresh and marine environments, while macroalgae only marine environments. There are many freshwater seaweeds as well.
Line 44: “ has its own contribution”, this implies each species contributes differently. I think it varies at a larger group level and not at a species level.
In general, I think the current introduction does not contribute to the overall intention of the manuscript. Instead, is a little misleading providing information about seaweeds and microalgae that are out of scope. For example, providing a general description of the morphology of algae is irrelevant here as it makes no difference in seaweeds ability to function as biofilters.
Something similar occurs with the seaweed composition section: Yes, the section is informative but not relevant to have its own section.
I will suggest reworking these sections and focusing on describing how wastewaters are treated nowadays, including, challenges, costs, capacities to do so and so on. After a thorough description, I will then include why seaweeds could be a great option of massive or small wastewater treatments. By doing so, your title, abstract, and introduction sections will be totally linked and you can proceed with the other sections that deal with different categories of pollutants.
Tables:
A great percentage of the species names are missing a space between the genus and the specific descriptor. Also, the captions revision as there seems to be words missing to clearly convey the message.
3rd Column : Do you mean wastewater source?
4th column: Is not clear to me the difference between biological, biofiltration, and adsorption. All of them can be done by seaweeds.
5th column: treatment? Do you mean what is removed?
There are some recent and fairly recent papers that could contribute to strengthen your point on how seaweeds could aid treating waters, particularly at non-focal scales. Kim et al 2015 published a couple of paper from work done in Long Island Sound, NY. Also, Buschmann et al 2018 have papers on nutrients bio extraction close to Salmon farms in Chile.